# How Do Plants Respond Biochemically to Fire? The Role of Photosynthetic Pigments and Secondary Metabolites in the Post-Fire Resprouting Response

Ana Carolina Santacruz-García [1],*, Sandra Bravo [2], Florencia del Corro [2], Elisa Mariana García [1], Domingo M. Molina-Terrén [3] and Mónica Azucena Nazareno [1]

[1] Consejo Nacional de Investigaciones Científicas y Técnicas CONICET and Instituto de Ciencias Químicas, Facultad de Agronomíay Agroindustrias, Universidad Nacional de Santiago del Estero, Santiago del Estero C.P. 4200, Argentina; marian_sgo@yahoo.com.ar (E.M.G.); manazar2004@yahoo.com (M.A.N.)

[2] Facultad de Ciencias Forestales, Instituto de Silvicultura y Manejo de Bosques INSIMA, Universidad Nacional de Santiago del Estero, Santiago del Estero C.P. 4200, Argentina; sandrabrav@gmail.com (S.B.); florenciadelcorro@gmail.com (F.d.C.)

[3] Facultad de Ciencias e Ingeniería Agroalimentarias y Forestales, Universidad de Lleida, Av. Rovira Roure 191, CP 25198 Lleida, Spain; dmolinat@gmail.com

* Correspondence: anacaro.santacruz@gmail.com

**Abstract:** Resprouting is one of the main regeneration strategies in woody plants that allows post-fire vegetation recovery. However, the stress produced by fires promotes the biosynthesis of compounds which could affect the post-fire resprouting, and this approach has been poorly evaluated in fire ecology. In this study, we evaluate the changes in the concentration of chlorophylls, carotenoids, phenolic compounds, and tannins as a result of experimental burns (EB). We asked whether this biochemical response to fire could influence the resprouting responses. For that, we conducted three EB in three successive years in three different experimental units. Specifically, we selected six woody species from the Chaco region, and we analyzed their biochemical responses to EB. We used spectrophotometric methods to quantify the metabolites, and morphological variables to estimate the resprouting responses. Applying a multivariate analysis, we built an index to estimate the biochemical response to fire to EB per each species. Our results demonstrate that photosynthetic pigment concentration did not vary significantly in burnt plants that resprout in response to EB, whereas concentrations of secondary metabolites (phenolic compounds and tannins) increased up to two years after EB. Our main results showed that phenolic compounds could play a significant role in the resprouting responses, while photosynthetic pigments seem to have a minor but significant role. Such results were reaffirmed by the significant correlation between the biochemical response to fire and both resprouting capacity and resprouting growth. However, we observed that the biochemical response effect on resprouting was lower in tree species than in shrubby species. Our study contributes to the understanding of the biochemical responses that are involved in the post-fire vegetation recovery.

**Keywords:** fire ecology; fire response; photosynthetic pigments; phenolic compounds; resprouting; secondary metabolites; vegetation recovery

## 1. Introduction

Fire has always played an important ecological role in the shaping of different regions [1]. In addition, it also plays a key role in the conservation and management of biodiversity in regions that have coevolved with this disturbance (e.g., Mediterranean ecosystems). This scenario has been described in semiarid landscapes, where seasonality and vegetation characteristics promote the fire occurrence [2,3]. However, within the global context of climatic and land-use changes, fire regimes have been altered, increasing in

severity and extension [3]. The restoration of the original plant community in post-fire environments will depend on the balance between the species regeneration strategies, which in turn are closely related to the pattern of disturbances [4–7]. Thus, plant strategies for tolerance to environmental disturbances should be studied as a challenge within this current scenario.

Post-fire vegetation regeneration strategies include resprouting ability from bud banks and, or germination through seed banks [7]. In post-fire environments, resprouting represents a primary persistence mechanism [5]. In particular, resprouting has been recognized as the main regeneration strategy for woody species in our study area [3,8]. This strategy is conditioned by the fire severity, which depletes the bud bank, their protection and, resourcing of regrowth [7].

However, the biochemical mechanisms involved in the post-fire resprouting have scarcely been studied. Plants are biochemically prepared to respond to environmental changes through the biosynthesis of secondary metabolites. Indeed, physical stress produced by fire promotes the biosynthesis of these compounds (e.g., terpenoids, phenolic compounds [9]). This enhanced biosynthesis provides resistance to vegetation and predisposes it to new fire events, which could be even more severe. Recent studies have highlighted the effect of foliar organic chemistry on flammability, focusing on volatile organic compounds (VOCs) such as terpenoids, which are characterized by reducing ignition temperatures in both foliage and litter [9–13] Under these conditions, the biosynthesis of secondary compounds could be considered as an "exaptation", as it is related to plant traits with specific functions that indirectly increase plant flammability [11,14].

The biochemical process associated with the biosynthesis of chlorophylls, carotenoids, tannins, and phenolic compounds involved in response to fire have still been poorly investigated. These compounds are characterized by having important regulatory functions on plants. For example, chlorophyll is one of the main indicators of the photosynthetic capacity and physiological status of plants [15–17]. Besides, carotenoids are considered important elements of information storage in response to environmental changes, whose function, beyond being an accessory pigment in photosynthesis is the protection of chlorophylls against photosensitization [18–20]. The study of the photosynthetic pigments is important due to under stress conditions, their quantification contributes to determining the plant behavior [15,16,21–23]. Studies on plant biochemical response to fire should consider the foliar persistence which could influence the total content of compounds due to the presence or absence of leaves during winter months [24,25].

Additionally, within secondary metabolites, phenolic compounds are considered as stress bioindicators due to their high sensitivity to changes in environmental conditions [13,26,27]. Indeed, Cannac et al. reported an increase in the synthesis of phenolic compounds in response to prescribed burnings in Pinus laricio (*Pinus nigra* ssp *laricio* (Poir.) Maire var. *corsicana* (Loud.) Hyl.). In addition to the above-mentioned functions, tannin biosynthesis is considered among the most important plant defense mechanisms against herbivores [28]. Moreover, phenolic compounds are recognized as strong antioxidant agents that protect cells against free radical damage [29,30].

This work proposed a novel approach of the resprouting since a biochemical understanding. A positive relationship between the biochemical response to fire and resprouting could contribute to elucidate the role of bioactive compounds in the post-fire vegetation recovery, where these compounds could act as drivers of resprouting [7]. Photosynthetic pigments are involved in the accumulation of carbohydrate reserves [31,32] Additionally, secondary metabolites contribute to the protection of the newly formed structures in the post-fire environment [7,33,34]. Thus, the biochemical response could be considered as a bioindicator or biomarker of the impact of the experimental burns [35].

The Chaco Region forests have been strongly affected by anthropic disturbances such as wildfires, livestock, logging and mechanical treatments to silvopastoral systems [33,36]. Forest management strategies include tools such as mechanical treatments and prescribed burning to shrub clearing and to improve pastures for livestock [25,37–39]. However, the

response of native vegetation to this disturbance possibly reveals certain adaptation and tolerance to fire [11,36]. For example, native woody species of the Argentine Chaco region are characterized by having a fast regeneration by resprouting [34,37,40]. The knowledge of the effect of plant biochemical response to fire contributes to understanding the patterns of postfire recovery of native vegetation in burnt forests. Besides, it could provide tools to post-fire management strategies [9,12,35,41,42].

In this study, we aimed (a) to evaluate the effect of experimental burns in the concentrations of chlorophylls, carotenoids, phenolic compounds, and tannins in leaves of six woody species, (b) to determine the temporal dynamics of the biosynthesis of these compounds in response to fire, and (c) to establish the link between plant biochemical response to fire with the resprouting responses. Here, we test the hypothesis that concentrations of photosynthetic pigments and secondary metabolites in the selected species change in response to experimental burns as a biochemical response to fire. These changes are related to the plant defense against the physical stress, the post-fire defoliation, and the time elapsed after the disturbance [9,11,12,17,35,43,44]. We proposed that this response is positively related to the post-fire resprouting [7,45,46].

This study represents the first effort to evaluate the plant responses to fire by using a strict biochemical approach. Indeed, most studies evaluate response to fire without considering ecophysiological behavior. The knowledge of plant biochemical response to fire could contribute to understanding the mechanisms of plant tolerance and resilience in natural environments. A temporary variation on the concentration of chlorophylls and secondary metabolites in response to fire could reveal the role of these compounds in the resprouting response, and their link to the plant tolerance to fire.

## 2. Materials and Methods

### 2.1. Experimental Site

The study area was located in the Western Chaco Region, Argentina. The sampling sites were located in the Experimental station "Francisco Cantos", Santiago del Estero, Argentina (28°03′ S, 64°15′ E). The climate is seasonal semiarid. The mean temperature was 20.6 °C and the mean annual precipitation was 731.2 mm during the studied years (2016–2018) (INTA climatic record, 2000–2018 annual series). Soils are regosols [47].

Typical vegetation includes a mosaic of forest, grasslands, shrublands, and savannas. The native forest is distributed in three strata; the dominant tree species of the upper stratum are quebracho-blanco (*Aspidosperma quebracho-blanco* Schltdl.) and quebracho colorado (*Schinopsis lorentzii* (Griseb.) Engl.), reaching over 20 m in height. The medium stratum includes mistol (*Sarcomphalus mistol* (Griseb.) Hauenschild, reaching from 7 to 12 m in height. In the understory, thorny species as atamisquí (*Atamisquea emarginata* Miers Ex Hook. & Arn), tala (*Celtis ehrenbergiana* (Klotzsch) Liebm), and molle (*Schinus johnstonii* F.A. Barkley), are common.

Fire has been an ecological event of the Chaco Region since the late 19th century [48,49]. Since the last century, the Chaco vegetation has experienced changes in the land-use as well as anthropogenic disturbances, which altered its fire regime. The fire season in the Chaco region coincides with the long dry and cold season, which extends from May to October. Along this period, the variation in both moisture content and phenological state of vegetation affects the plant flammability [25].

The experiment was conducted on a typical forest of this region characterized by wildfires and mechanical treatments to control shrub encroachment and to improve the pasture growth for livestock [25,37–39]. The area where we established the plots correspond to forests in which accidental fires of unknown dates have occurred. Scars and other signs such as wood debris and charred bark suggest a recent one (less than 10 years ago). Six woody species were selected according to their representativeness in the three forest strata within the study area [50]. Botanical family, growth habit, and foliar persistence were considered in the selection of the species, as these traits were intended to use in the analysis

and the subsequent discussion (Table 1). Names of species follow the nomenclature system devised by the Instituto de Botánica Darwinion (Buenos Aires, Argentina).

**Table 1.** Botanical family, foliar persistence, and growth habit for the studied species.

| Species | Botanical Family | Growth Habit | Foliar Persistence |
|---|---|---|---|
| *A. emarginata* | Capparaceae | *Shrub* | Evergreen |
| *A. quebracho blanco* | Apocynaceae | *Tree* | Evergreen |
| *C. ehrenbergiana* | Celtidaceae | *Shrub* | Deciduous |
| *S. johnstonii* | Anacardiaceae | *Shrub* | Evergreen |
| *S. lorentzii* | Anacardiaceae | *Tree* | Deciduous |
| *S. mistol* | Rhamnaceae | *Tree* | Deciduous |

*2.2. Characterization of Experimental Burns*

To evaluate the plant response to fire, three experimental burns (EB) were carried out in three successive years (2016–2018). In the sampling site, 170 individual plots of 2 m × 2 m were randomly established. Sixty plots were burnt each year, with ten replications for each species. Individuals of *A. emarginata* were not burnt in 2018 because of their absence in the plots destined to burn during that year. In the center of each plot, we tagged an individual plant of each species (DBH < 15 cm, and pre-fire height < 4 m). A low fine fuel load was used, 4000 kg DM ha$^{-1}$, corresponding to the aboveground biomass in native grass in the Chaco region [33,36,38]. In plots where the fine fuel load did not reach the expected amount, a previously weighed dry grass biomass was homogeneously distributed.

EB were conducted at the end of the fire season each year (September–October) which corresponds to the highest flammability level observed in the Chaco region [25,33]. Meteorological conditions during EB dates were monitored every 30 min (air temperature(°C), air relative humidity (%), wind speed (km h$^{-1}$), and wind direction) for the maintenance of acceptable thresholds of fire behavior (flame length < 3 m). A drip-torches were used to lit a head fire. As a fire severity measurement, burnt biomass (BB, %) was estimated visually by two operators and then averaged to reduce the experimental error. As a fire intensity measurement, the burning time of each plant (BT, s) and flame length (FL, m) were recorded [51]. The BT of each plant was considered as the time duration from ignition until flame extinguishment, and FL was registered as the visual estimation of two independent observers. We considered that the greater BB, the greater fire severity, whereas, the longer BT and FL, the greater fire intensity (Table 2) [33].

**Table 2.** Mean and standard deviation for the pre-fire height (m), burning time (BT, s), flame length (FL, m), and burnt biomass (BB, %). Different letters indicate significant differences, according to LSD Fisher pairwise comparison procedure with α: 0.05.

| Species | Pre-Fire Height | BT | FL | BB |
|---|---|---|---|---|
| *A. emarginata* | 1.88 ± 0.56 [a] | 62 ± 19 | 1.26 ± 0.19 | 58 ± 20 [c] |
| *A. quebracho blanco* | 1.92 ± 0.61 [a] | 65 ± 31 | 1.37 ± 0.18 | 30 ± 19 [ab] |
| *C. ehrenbergiana* | 1.66 ± 0.54 [a] | 77 ± 41 | 1.24 ± 0.21 | 40 ± 26 [bc] |
| *S. johnstonii* | 1.37 ± 0.22 [a] | 63 ± 32 | 1.28 ± 0.29 | 58 ± 19 [c] |
| *S. lorentzii* | 3.96 ± 1.89 [b] | 64 ± 41 | 1.29 ± 0.21 | 24 ± 29 [ab] |
| *S. mistol* | 3.96 ± 1.34 [b] | 102 ± 52 | 1.22 ± 0.17 | 19 ± 16 [a] |

*2.3. Assessment of Chlorophyll and Carotenoid Contents*

Chlorophyll and carotenoid contents were determined in burnt plants from EB 2016, as a preliminary approach to evaluate the potential plant response to fire. Samples of five

burnt individuals of each species were collected six and twelve months after EB 2016. As a control, samples of five unburnt individuals of each species were randomly collected from the same site on both dates (6 and 12 months after the EB date). During the trial time period, the species went through different phenological states: (a) six months after EB (April 2017, the autumn season in Southern hemisphere), deciduous species shed their foliage, but evergreen species could maintain their leaves and continue growing; (b) twelve months after EB (October 2017, springtime season in Southern hemisphere), the species were beginning their active growth or were already actively growing.

Representative leaf samples of each individual plant species (0.6 g of dried material) were extracted with 50 mL of acetone using a blender (Pro Scientific, Oxford, MS, USA) and filtered under vacuum. This procedure was repeated twice, and these three extracts were combined and then transferred to diethyl ether by adding small portions of the acetone extract and large amounts of water in a separatory funnel. The ethereal extract was separated and taken to a 10 mL final volume. Extracts were prepared in duplicate. Determinations were performed using a UV/vis spectrophotometer (7315 Spectrophotometer, Jenway, Staffordshire, UK), and absorbances were measured at 430, 642, and 660 nm. Comar and Zscheile equations were used to calculate chlorophylls a, b, and total contents [52]. Total carotenoid contents were calculated using the carotenoid absorption coefficients [52].

### 2.4. Total Phenolic Compounds and Tannin Assessments

To analyze the effect of temporal variations on the total contents of phenolic compounds and tannins, leaf samples from five individual plants of each species burnt during each EB (2016, 2017, and 2018) were collected in November 2018 for this experiment. In November 2018, the closest test date to EB, the leaves of all studied species were fully expanded for sampling. As a control, samples of five individual plants of each species were collected in an unburnt plot of the same study area. The results correspond to different individual plants collected simultaneously in November 2018 to ensure that all plants were exposed to the same environmental conditions before analysis. Every plant was burnt once, and no one was treated twice.

Representative samples of each individual plant (0.1 g of dried material) were extracted with 10 mL of acetone/water (70/30, v/v) in an ultrasonic bath (HH-S Water Baths, Bioamerican Science, Buenos Aires, Argentina) for 60 min, at room temperature. Extracts were centrifuged at 10,000 rpm for 10 min at room temperature. The supernatant completed to 10 mL with the acetone/water mixture (70/30, v/v) was used to the assessments of phenolic compound and tannin contents according to the methodology described in García (2015).

The Folin–Ciocalteu method was used to determine the total contents of phenolic compounds. For calibration purposes, gallic acid (GA) was taken as a reference chemical standard. GA solutions (0.05, 0.06, 0.10, and 0.15 mg mL$^{-1}$) were prepared in distilled water. An aliquot of 0.15 mL of each extract was transferred to a 5 mL volumetric flask containing distilled water (1.05 mL) and Folin-Ciocalteu reagent (0.65 mL). The volume was completed with 3.15 mL of anhydrous sodium carbonate (20% w/v). The mixture was stirred and heated at 50 °C for 10 min in the same conditions as the blank without plant extract. The absorbance of the samples was measured at 725 nm in a UV/Vis spectrophotometer. Results were calculated and expressed as GA equivalents 1000 mg$^{-1}$ of the sample [28].

For extractable condensed tannins (ECT) determinations, the butanol/HCl assay was used. In a volumetric flask, 3 mL of butanol/hydrochloric acid (95/5, v/v) was added to 0.5 mL of each extract. Each mixture was stirred and heated in a water bath at 95 °C for 60 min, and then, it was cooled in an ice bath up to reaching room temperature. Afterward, sample absorbance was measured at 550 nm. ECT were calculated using the tannin absorption coefficients [52] and results were expressed as g of cyanidin-3-glucoside 100 g$^{-1}$ dry weight [28].

### 2.5. Resprouting Measurements

To determine the growth of resprouts and post-fire resprouting capacity (RC), ten individual plants per species were georeferenced and assessed before EB 2016 (October). Resprouting measurements were performed during the growing season following the treatment when new growth modules were fully expanded (March–April). The complete loss of foliage and absence of sprouts (green shoots) was considered as an indicator of mortality. In alive plants, the number of resprouts, the basal diameter of each resprout and the maximum length of resprouts were recorded. Those variables were considered as indicators of the resprout growth in our study. Resprouting capacity per species was assessed by the multiplication of the mean of burnt biomass and the percentage of individual plants that have resprouted and divided it by 100 to keep the results in a range from 0 to 100 (unitless) [53].

### 2.6. Statistical Analysis

For assessments of chlorophyll a, chlorophyll b, total chlorophyll, and carotenoids contents, data were analyzed through a mixed linear model (MM), using species, fire data (6 or 12 months after EB), and treatment (Burnt or Unburnt) as fixed effects. The individual plant was considered a random effect. For assessments of total phenolic compounds and tannin contents, a generalized linear mixed model (GLMM) was also performed using Normal distribution and species and EB year as fixed effects. The individual plant was considered a random effect.

Regarding resprouting and their link to the biochemical response to fire, we considered the data from the burnt plants in EB 2016, due to their coincidence with the resprouting measurements. An ANOVA was performed to examine whether the differences observed in burning time, flame length, burnt biomass, and growth of resprouts (number, diameter, and height) were statistically significant or not. We used the same analysis to understand whether species and growth habit had a significant effect on resprouting capacity and the growth of resprouts. To evaluate the photosynthetic compounds and secondary metabolites' effect over resprouting capacity, we performed a backward regression, whereas to evaluate the same effect over resprouting growth (number, diameter, and height) we used a general linear model (GLM). We performed a principal component analysis (PCA) to determine the plant's biochemical response to fire value. The variables considered in the PCA included the total contents of the biochemical compounds evaluated in this work for the burnt plants (from EB 2016). The first axis of the PCA was used to obtain a biochemical response index per each species. The correlation between resprouting capacity and the biochemical response was performed through a Spearman's correlation coefficient. The same correlation was used between number, diameter, and a maximum height of resprouts with the biochemical response to fire index. The statistical software used was Infostat/2017 (InfoStat Group V.2017, Córdoba, Argentina) with an $\alpha = 0.05$.

## 3. Results

### 3.1. Assessment of Chlorophyll and Carotenoid Contents

A wide range of total chlorophyll contents among the different species and treatments were observed, varying from $494.9 \pm 333.9$ (burnt plants, six months after EB, *A. emarginata*) to $1516.7 \pm 102.2$ (control plants, twelve months after EB, *C. ehrenbergiana*) $\mu$g g$^{-1}$ dry weight. A non-significant effect of EB was observed in the concentration of this pigment in burnt plants with respect to control (F = 1.53, $p = 0.2191$). A non-significant decrease in the content of this pigment was observed in *S. lorentzii*, *C. ehrenbergiana*, and *S. mistol* in burnt plants with respect to control. In contrast, burnt plants of *S. johnstonii* and *A. emarginata* showed a non-significant increase in the concentration of this pigment (Figure 1A).

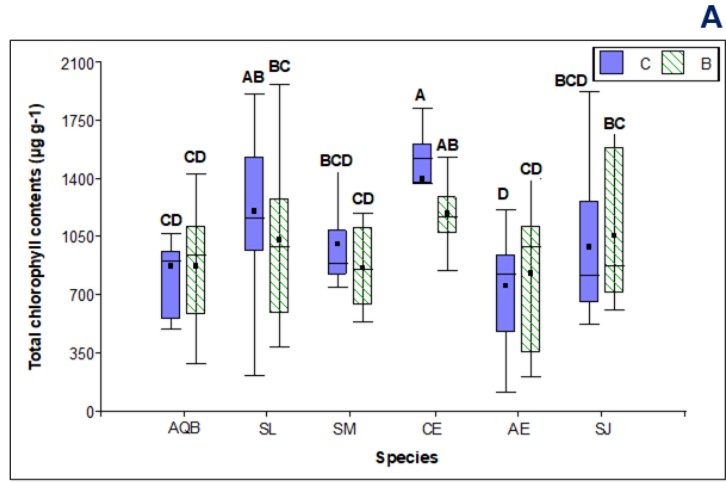

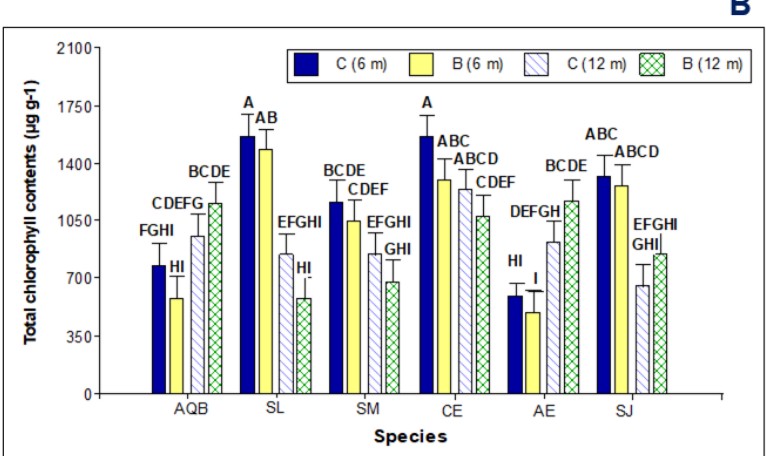

**Figure 1.** (**A**) Content of total chlorophyll in response to experimental burns (EB) expressed in µg g$^{-1}$ dry weight. A black line within the box marks the median, a black point within the box marks the mean, and the boundary of the box farthest from zero indicates the third quartile (Q3). Whiskers above and below the box indicate the 10th and 90th percentiles. (**B**) Comparison of total chlorophyll contents considering the time elapsed after EB 2016 (6 and 12 months). Different letters indicate significant differences, according to LSD Fisher pairwise comparison procedure with α: 0.05, n = 30 individual plants by treatment each sampling date. In each figure (B) = Burnt plants; (C) = Control plants. References: SM = *S. mistol*, SJ = *S. johnstonii*, SL = *S. lorentzii*, AE = *A. emarginata*, AQB = *A. quebracho-blanco*, CE = *C. ehrenbergiana*.

A significant effect of the time elapsed after EB on the chlorophyll biosynthesis was observed in the studied species (F = 11.97, *p* = 0.0008). Six months after EB, each of the six studied species have shown a non-significant decrease in the total content of chlorophyll in burnt plants with respect to control. Twelve months after EB, burnt plants of deciduous species (*S. lorentzii*, *C. ehrenbergiana*, and *S. mistol*) have indicated a decrease in the concentration of this pigment compared to the control. Burnt plants of evergreen species (*A. quebracho-blanco*, *A. emarginata*, and *S. johnstonii*) have shown an increase in these contents compared to the control plants in the same period (Figure 1B).

Results did not indicate a significant effect of the EB in the contents of chlorophyll *a* (Chl *a*) and *b* (Chl *b*) among burnt plants and control (F = 2.19, *p* = 0.1421; F = 0.35, *p* = 0.5579; Chl *a* and Chl *b*, respectively). When the measurement dates were compared, results indicated a significant decrease in the chlorophyll *a* contents twelve months after EB compared with samples measured six months before (F = 30.02, *p* < 0.0001). Neither of the dates of measurement indicated significant differences in the chlorophyll *b* contents of the

samples treated (F = 3.70, *p* = 0.0576). A significant effect of the species in the concentration of these pigments was observed (F = 8.95, *p* < 0.0001; F = 12.39, *p* < 0.0001; Chl *a* and Chl *b*, respectively).

The mean concentration of total carotenoids varied from 124.6 ± 73.9 (burnt plants, six months after EB, *A. emarginata*) to 611.1 ± 244.2 (control plants, twelve months after EB, *C. ehrenbergiana*) µg g$^{-1}$ dry weight. The studied species have shown non-significant differences in the total content of carotenoids in burnt plants compared to control, except to *C. ehrenbergiana* (F = 2.44, *p* = 0.1214). The total carotenoid contents have decreased in burnt plants with respect to control in *A. quebracho-blanco, S. lorentzii, S. mistol,* and *C. ehrenbergiana*. This decrease was significant only for the last species. *S. johnstonii* and *A. emarginata* showed a non-significant increase in the concentration of this pigment in burnt plants compared to the control (Figure 2A).

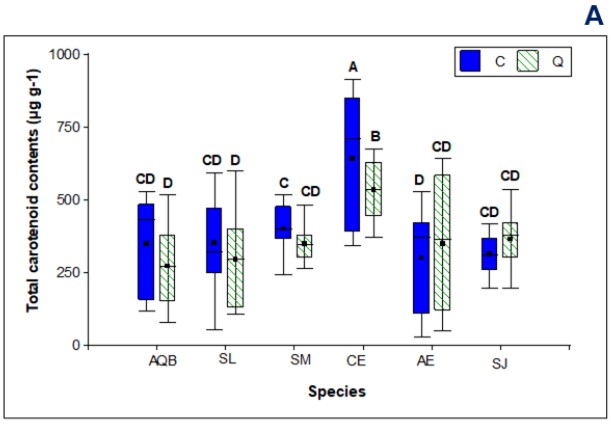

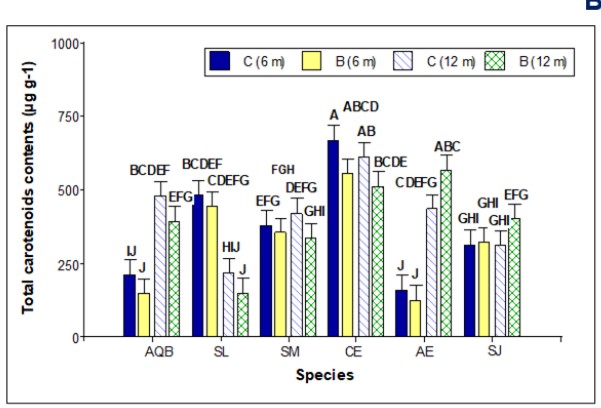

**Figure 2.** (**A**) Content of total carotenoid in response to EB expressed in µg g$^{-1}$ dry weight. In the box plot, the boundary of the box closest to zero indicates the first quartile (Q1), a black line within the box marks the median, a black point within the box marks the mean, and the boundary of the box farthest from zero indicates the third quartile (Q3). Whiskers above and below the box indicate the 10th and 90th percentiles. (**B**) Comparison of total carotenoid contents considering the time elapsed after EB 2016 (6 and 12 months). Different letters indicate significant differences, according to LSD Fisher pairwise comparison procedure with α: 0.05, n = 30 individual plants by treatment each sampling date. In each figure (B) = Burnt plants; (C) = Control plants. References: SM = *S. mistol*, SJ = *S. johnstonii*, SL = *S. lorentzii*, AE = *A. emarginata*, AQB = *A. quebracho-blanco*, CE = *C. ehrenbergia.na*.

Results indicated a significant effect of the time elapsed after EB in the total carotenoids contents (F = 7.65, *p* = 0.0069). Six months after EB, burnt plants of the studied species (except to *S. johnstonii*), have shown a non-significant decrease in the concentration of this pigment in burnt plants compared to the control. Twelve months after EB, burnt plants

of deciduous species (*S. lorentzii*, *C. ehrenbergiana*, and *S. mistol*) revealed a decrease in the total content of this pigment compared to the control values. Whereas, burnt plants of evergreen species (*A. emarginata* and *S. johnstonii*) have shown a considerable increase in the total contents of these pigments with respect to control in the same period (Figure 2B).

### 3.2. Total Phenolic Compound and Tannin Assessments

The mean concentration of total phenolic compounds (TPC) in the leaves varied from $6.2 \pm 1.2$ (EB 2017, *A. emarginata*) to $480.9 \pm 178.8$ (EB 2017, *S. mistol*) mg of gallic acid per 1000 mg of sample. A significant increase in the concentration of TPC was observed in burnt plants with respect to control (F = 4.06, $p < 0.0001$). The increase of TPC was observed for two months after EB in new leaves fully expanded in resprouts post-fire, to twenty-six months after EB in leaves fully expanded. These contents have not returned to normal values 26 months later (Figure 3A). All studied species have shown significant differences in TPC concentration (F = 122.8, $p < 0.0001$). The species *S. lorentzii* and *S. johnstonii* have shown the highest concentrations of this metabolic group, whereas *A. emarginata* showed the lowest concentration (Figure 3B). However, a tendency related to foliar persistence was not observed.

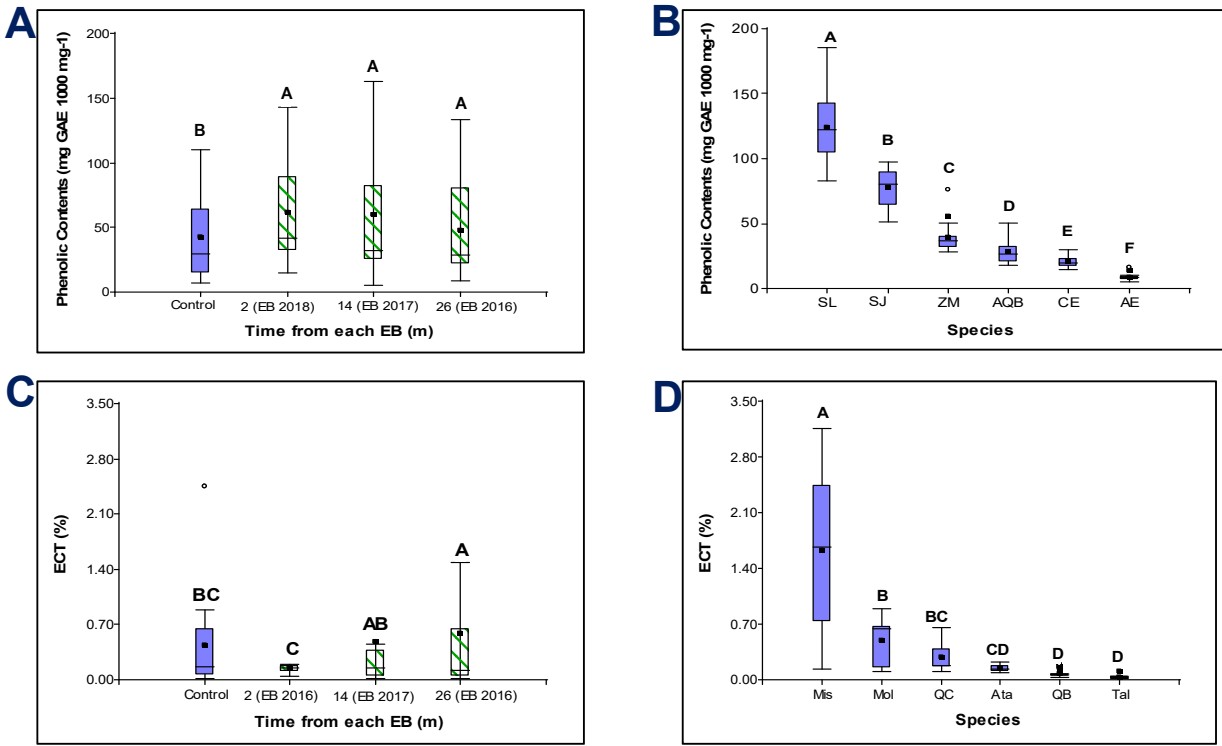

**Figure 3.** Content of total phenolic compounds (mg of gallic acid 1000 mg$^{-1}$ of sample) in response to three experimental burns (EB) from 2016–2018 (**A**); measured in leaves of six woody species from the Western Chaco region, Argentina, collected during November 2018 (**B**). Samples of five individual plants were burnt in three different experimental burns (EB). Concentration of extractable condensed tannin (ECT, %) in response to three experimental burns (EB) from 2016–2018 (**C**); measured in leaves of six woody species from the Western Chaco region, Argentina, collected during November 2018 (**D**). Samples of five individual plants were burnt in three different experimental burns (EB). Each EB was performed at the end of fire season (September or October) in a different year (2016, 2017, and 2018). Any plant was burnt twice. In the box plot, the boundary of the box closest to zero indicates the first quartile (Q1), a black line within the box marks the median, a black point within the box marks the mean, and the boundary of the box farthest from zero indicates the third quartile (Q3). Whiskers above and below the box indicate the 10th and 90th percentiles. Different letters indicate significant differences between sampling dates, according to LSD Fisher pairwise comparison procedure with α: 0.05, n = 30 individual plants in each EB and 30 individual plants as control. References: SL = *S. lorentzii*, SJ = *S. johnstonii*, SM = *S. mistol*, AQB = *A. quebracho-blanco*, CE = *C. ehrenbergiana*, AE = *A. emarginata*.

The mean concentration of extractable condensed tannins (ECT) varied from $0.02 \pm 0.01$ (EB 2016 and EB 2017, *C. ehrenbergiana*) to $2.36 \pm 0.89$ (EB 2016, *S. mistol*) g of cyanidin-3-glucoside 100 $g^{-1}$ dry weight. ECT concentration in burnt plants varied significantly from control plants (F = 10.44, $p < 0.0001$). The results have shown a non-significant decrease in ECT contents, two months after EB, as compared with control plants. Fourteen months after EB, ECT concentrations have not shown significant differences among burnt and control plants, whereas, at twenty-six months after EB, ECT contents in burnt plants have shown significant increase with respect to control (Figure 3C). *S. mistol* have shown the highest ECT concentrations, whereas *A. quebracho-blanco* and *C. ehrenbergiana* showed the lowest concentration of this secondary metabolite (F = 22.79, $p < 0.0001$; Figure 3D). The results were not related to the foliar persistence of the studied species and, CV showed a high intra- and interspecific variation in TPC and ECT contents (Figure 3B,D).

### 3.3. Resprouting Measurements

Table 3 showed the pre-fire height, the mortality (%), the resprouting percentage (percentage of plant population that has resprouted), and the resprouting capacity (RC) of each species. The mean pre-fire height for shrubby and tree species was $1.63 \pm 0.48$ m and $3.28 \pm 1.62$ m, respectively (F = 14.11, $p = 0.0008$). There was no post-fire mortality in plants of any species. Most of the studied species had basal resprouts and were multi-stemmed after EB, except for *S. mistol*, which was the only species that did not resprout after fire. Even with a high standard deviation, the number of resprouts was similar among the studied species ($11 \pm 11$, F = 2.17, $p = 0.0913$), and the maximum height of resprouts was not significantly different among species ($0.52 \pm 0.30$ m, F = 1.72, $p = 0.1854$). Instead, the diameter of resprouts varied among species (F = 8.07, $p = 0.005$).

**Table 3.** Mean and standard deviation for the post-fire resprout number (RN), resprout diameter (RD, cm), and resprout maximum height (RMH, m). Additionally, this table showed the type of resprout, the post-fire mortality (%), the resprouting percentage (percentage of plant population that have resprouted, RP), and the resprouting capacity (RC) of each species. Resprouting measurements were made with data from EB 2016. Different letters indicate significant differences, according to LSD Fisher pairwise comparison procedure with α: 0.05.

| Species | Type of Resprout | Mortality | RN | RD | RMH | RP | RC |
|---|---|---|---|---|---|---|---|
| *A. emarginata* | Basal | 0 | $21 \pm 14$ | $0.21 \pm 0.13$ [a] | $0.43 \pm 0.31$ | 80 | 46.4 |
| *A. quebracho blanco* | Basal | 0 | $8 \pm 5$ | $2.30 \pm 1.60$ [b] | $0.46 \pm 0.29$ | 90 | 27.0 |
| *C. ehrenbergiana* | Basal | 0 | $11 \pm 12$ | $0.32 \pm 0.06$ [a] | $0.66 \pm 0.54$ | 70 | 28.3 |
| *S. johnstonii* | Basal | 0 | $14 \pm 8$ | $0.35 \pm 0.10$ [a] | $0.74 \pm 0.30$ | 100 | 58.5 |
| *S. lorentzii* | Basal | 0 | $6 \pm 11$ | $0.16 \pm 0.17$ [a] | $0.33 \pm 0.30$ | 40 | 9.6 |
| *S. mistol* | - | 0 | $0 \pm 0$ | - | - | 0 | 0.0 |

Burning time and flame length showed non-significant differences among the studied species ($72 \pm 39$ s, F = 1.78, $p = 0.1334$, and $1.28 \pm 0.21$ m, F = 0.63, $p = 0.6803$, respectively). However, results indicated significant differences in burnt biomass among studied species (F = 5.97, $p = 0.0002$). Indeed, the amount of burnt biomass of shrubby species was higher than that of the tree species. Shrubby species showed both a higher resprouting percentage and a higher resprouting capacity than that of tree species. The RC in *A. quebracho-blanco* was especially higher than in the other tree species, and the RC in *S. johnstonii* was slightly higher than in the other shrubby species (Table 3).

Table 4 showed that both the maximum height of resprouting and diameter of resprouts were significantly affected by the species (F = 4.96, $p = 0.0029$, and F = 8.60, $p = 0.0001$, respectively), while growth habit showed a significant effect in all the studied parameters (maximum height of resprouts, diameter of resprouts, number of resprouts, and resprouting capacity).

**Table 4.** Results of ANOVA on the number of resprouts, maximum height of resprouts, diameter of resprouts, and resprouting capacity. The explanatory variables were in all cases species and growth habit. Values indicate the F-values and stars indicate the significance level. Significance levels: ** <0.01; * <0.05; <0.1.

|  | Number of Resprouts | Maximum Height of Resprouts | Diameter of Resprouts | Resprouting Capacity |
|---|---|---|---|---|
| Species | 0.96 | 2.93 * | 9.52 ** | 1.92 |
| Growth habit | 6.99 * | 13.10 ** | 4.88 * | 22.30 ** |

*3.4. Biochemical Compounds Affecting the Resprouting Capacity*

When analyzing the data without considering species differences, resprouting capacity was significantly affected by the chlorophyll *a* content, the total phenolic compounds, and tannin contents. However, we did not observe a significant effect of total chlorophyll content, chlorophyll *b* content, and total carotenoid content in resprouting capacity (Table 5).

**Table 5.** Effects of photosynthetic compounds and secondary metabolites over resprouting capacity in five native woody species from the Chaco Region. Values indicate the F-values of the backward regression between bioactive compounds and RC; stars indicate the significance level. Significance levels: ** <0.01; <0.1.

|  | Resprouting Capacity ($R^2 = 0.50$) |
|---|---|
| *Total chlorophylls* | 0.39 |
| *Chlorophyll a* | 13.69 ** |
| *Chlorophyll b* | 5.85 |
| *Total carotenoids* | 6.31 |
| *Total phenolic contents* | 9.87 ** |
| *Tannin contents* | 8.69 ** |

*3.5. Biochemical Compounds Affecting the Growth of Resprouts*

We observed different patterns across the studied species in relation to resprouting growth. For example, tree species showed a different response than shrubby species. Indeed, the last-mentioned species showed a higher effect of the bioactive compounds in the growth of resprouts (Table 6). The number of resprouts and the maximum height of resprouts did not differ significantly among the studied species (F = 2.17, *p* = 0.0913; and F = 1.72, *p* = 0.1854, respectively, Table 3). However, the diameter of resprouts differed among species (F = 8.07, *p* = 0.0005, Table 3).

Phenolic compounds were the most significant variable regulating the number of resprouts in most of the studied species, except for *S. lorentzii*. This number of resprouts in shrubby species *C. ehrenbergiana* and *S. johnstonii* seemed to be more affected by the bioactive compounds than the other species, due to the number of resprouts in those species was significantly affected by all the compounds tested, except for chlorophyll b in *S. johnstonii* (Table 6).

The relationship between the maximum height and the diameter of resprouts followed a different trend in shrubby and tree species. Phenolic compound levels did affect the diameter of resprouts in the tree species whereas the resprout height did not show a particular relationship with any variable. On the other hand, the maximum height and diameter of resprouts in shrubby species were positively associated with all the biochemical compounds studied (Table 6).

**Table 6.** Effects of photosynthetic compounds and secondary metabolites over number of resprouts, maximum height of resprouts, diameter of resprouts in five native woody species from the Chaco Region. Values indicate the F-values and stars indicate the significance level. Significance levels: *** <0.001; ** <0.01; * <0.05; <0.1.

| | Number of Resprouts | Maximum Height of Resprouts | Diameter of Resprouts |
|---|---|---|---|
| | *A. quebracho blanco* | | |
| Total chlorophylls | 6.01 | **7.86 *** | 5.82 |
| Chlorophyll a | 5.61 | 7.55 | 5.40 |
| Chlorophyll b | 7.26 | **8.59 *** | 7.21 |
| Total carotenoids | 6.38 | **8.19 *** | 6.09 |
| Total phenolic contents | **16.86 *** | **13.07 *** | **18.46 *** |
| Tannin contents | **9.24 *** | **11.89 *** | **9.48 *** |
| | *S. lorentzii* | | |
| Total chlorophylls | 3.63 | 6.26 | 4.24 |
| Chlorophyll a | 3.78 | 6.86 | 4.62 |
| Chlorophyll b | 2.49 | 4.24 | 3.02 |
| Total carotenoids | 3.80 | 5.58 | 3.69 |
| Total phenolic contents | 1.13 | 8.54 | **18.65 *** |
| Tannin contents | 2.69 | 6.83 | 6.25 |
| | *A. emarginata* | | |
| Total chlorophylls | 4.24 | **10.88 *** | **10.45 *** |
| Chlorophyll a | 4.03 | **10.93 *** | **10.27 *** |
| Chlorophyll b | 4.70 | **10.66 *** | **10.72 *** |
| Total carotenoids | 5.55 | **10.09 *** | **12.15 *** |
| Total phenolic contents | **14.22 *** | **9.56 *** | **22.54 ** ** |
| Tannin contents | 4.60 | **25.57 ** ** | **16.53 *** |
| | *C. ehrenbergiana* | | |
| Total chlorophylls | **20.15 *** | **36.15 ** ** | **69.32 ** ** |
| Chlorophyll a | **21.78 ** ** | **36.49 ** ** | **65.48 ** ** |
| Chlorophyll b | **17.37 *** | **34.47 ** ** | **72.95 ** ** |
| Total carotenoids | **11.07 *** | **19.69 *** ** | **39.19 ** ** |
| Total phenolic contents | **14.20 *** | **28.26 ** ** | **45.08 ** ** |
| Tannin contents | **11.72** | **45.20 ** ** | **182.90 ** ** |
| | *S. johnstonii* | | |
| Total chlorophylls | **8.92 *** | **14.13 *** | **17.74 *** |
| Chlorophyll a | **10.09 *** | **16.46 *** | **21.62 ** ** |
| Chlorophyll b | 6.48 | **9.65 *** | **11.18 *** |
| Total carotenoids | **12.21 *** | **16.14 *** | **21.38 ** ** |
| Total phenolic contents | **10.04 *** | **14.05 *** | **24.84 ** ** |
| Tannin contents | **13.82 *** | **31.29 ** ** | **50.41 ** ** |

### 3.6. Association between the Biochemical Response to Fire and Resprouting

The first two axes of the PCA performed to evaluate the plant biochemical response to fire explained 81.2% of the data variation. The first PCA axis (PC1) explained 47.3% of the variation in the data and was positively associated with photosynthetic pigments and negatively associated with both the concentration of extractable condensed tannins (ECT) and the total phenolic compounds (TPC). The second PCA axis (PC2) explained 33.9% of the variation in the data, which was mainly explained by the ECT (higher positive score on the PC2 axis) and by the total content of carotenoids (lower positive score on the PC2) (Figure 4A). PC1 was considered as the variable of the biochemical response to fire.

Spearman's rank correlation coefficient between resprouting capacity and the first component of the plant biochemical response showed a significant association between them (Spearman's $p = 0.83$, $p$-value < 0.0001; Figure 4B). Besides, this response showed a significant correlation with the number of resprouts (Spearman's $p = 0.53$, $p$-value = 0.0027; Figure 4C). The maximum height of resprouts (Spearman's $p = 0.61$, $p$-value = 0.0003; Figure 4D) and the diameter of resprouts (Spearman's $p = 0.51$, $p$-value = 0.0044; Figure 4E) were also positively correlated to the plant biochemical response to fire.

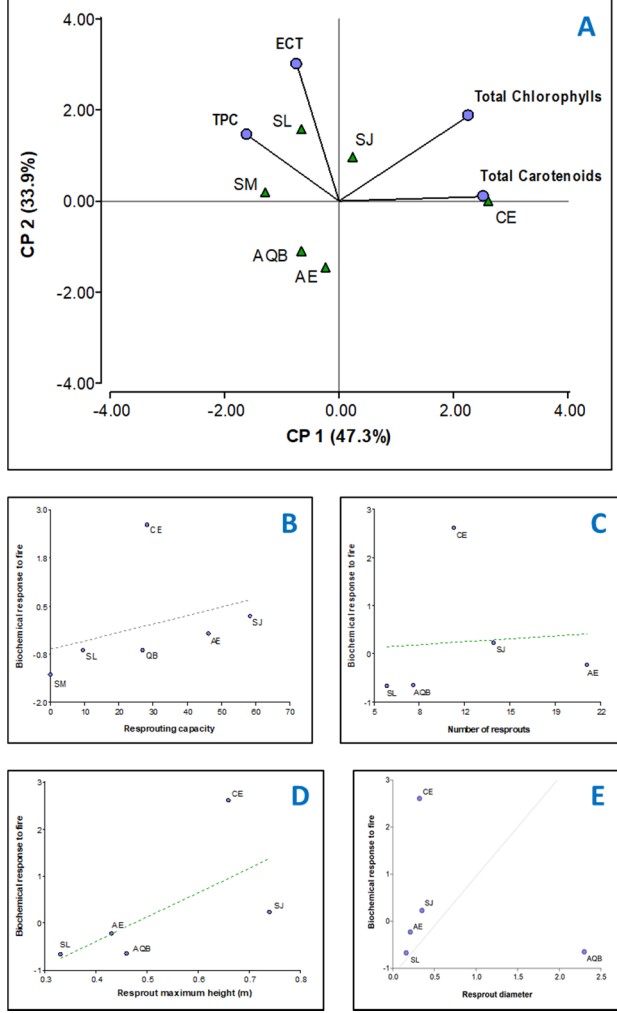

**Figure 4.** (**A**) Principal component analysis (PCA) of the plant biochemical response to fire. The ordination was based on the total contents of chlorophylls, carotenoids, total phenolic compounds (TPC), and extractable condensed tannins (ECT) from burnt plants in EB 2016. (**B**) Correlation between the resprouting capacity (RC), (**C**) the number of resprouts, (**D**) the resprouts maximum height, and (**E**) the resprouts diameter with the first component of the plant biochemical response to fire. The green line shows where the points would have a perfect correlation between RC and biochemical response to fire. Species below the line had a higher content of bioactive compounds in relation to RC, while species above the line had a higher response to fire compared to the content of bioactive compounds biosynthesized (Pearson's $p = 0.60$, $p$-value < 0.0001). References: SM = *S. mistol*, SJ = *S. johnstonii*, SL = *S. lorentzii*, AE = *A. emarginata*, AQB = *A. quebracho-blanco*, CE = *C. ehrenbergiana*.

## 4. Discussion

### 4.1. Assessing Chlorophyll and Carotenoid Contents

In contrast with our previous expectations, the total content of photosynthetic pigments (chlorophylls and carotenoids) was not significantly affected by fire. However, experimental burns seemed to affect the functional status of the woody species studied, at least in a short temporal distance from disturbance (six months). This could represent an indicator of plant biochemical tolerance to the physiological stress induced by fire [43,54–56]. Twelve months after EB, the increase of total chlorophyll content observed in evergreen species is in clear contrast to the decrease of this pigment in deciduous species. This fact suggests a differential response to fire according to the foliar persistence. This result is consistent with other studies reporting that evergreen and deciduous species have different strategies to protect photosynthetic pigments in stressful environments [24,25,57].

The lack of significant variations in Chl *a* and Chl *b* contents among control and burnt plants suggests a remarkable recovery of the plant physiological stability even six months after EB for the selected species. Additionally, the maintenance of chlorophyll *b* contents could indicate that vegetation has not demanded an increase in the concentration of this pigment, even at a short time after EB [15,17].

The relative stability of carotenoid contents after EB in most of the studied species could be related to the notable recovery of chlorophyll contents mentioned above, and a moderate antioxidant capacity of the carotenoids compared with the activity of phenolic compounds [58] These results seem to indicate both a strong resilience to fire of native vegetation and, or to an adaptation to climatic seasonality [18,19,34,59]. In our results, *C. ehrenbergiana* was the only species that showed a significant decrease in carotenoid contents in response to EB; however, this species showed the highest photosynthetic pigment contents among the studied species. These results could be related to the acquisitive strategy of this species in disturbed environments where it can act as a pioneer species [60].

The increase in the total content of carotenoids observed twelve months after EB in the evergreen species, *A. quebracho-blanco* and *A. emarginata* could represent the response of these species to stress conditions caused by the fire effect and the dryness period. These results reaffirm the role of carotenoids as antioxidants in response to environmental stress conditions beyond the physical stress produced by fire [19,61–65]. On the same date, the decrease in carotenoids contents observed in deciduous species may indicate that our results correspond to newly formed leaves that have not been affected by the fire and drought stress.

### 4.2. Total Phenolic Compounds and Tannin Assessments

In our work, the significant increase of phenolic compound contents observable even 26 months after EB could be attributed both to fire and to changes in postfire environmental conditions (high radiation levels, higher daily and seasonal thermal amplitude, and soil evaporation). Similar results were reported by Alonso et al. [66], which mentioned an increase in phenolic compounds in *Pinus pinaster* forests after the first months of thermal stress in bole and crown. Indeed, Cannac et al. considered these compounds as good stress bioindicators due to their high sensitivity to changing environmental conditions. Accordingly, the changes in the UV-radiation produce an increase in the biosynthesis of phenolic compounds [67] as a product of the stimulation of the plant antioxidant defenses to resist pathogens [13]. In favor of our results, there would be the fact that Bowman et al. [11] mentioned that this type of plant response to fire might be a mechanism of tolerance to disturbances that could also promote plant flammability.

The non-significant differences in tannin contents at two-months after EB could mean that the biosynthesis of these compounds is not affected by fire injury. However, the increasing tannin concentrations even two years after EB could be due to a greater need for defense against herbivores and pathogens in the post-fire environmental conditions [28]. This indirect response of vegetation to fire could be an exaptation as a result of other causes than fire, as the herbivory pressure [11]. For example, Jaureguiberry and Díaz [14] affirmed that in the Chaco Region, adaptations against large herbivores could furthermore provide fire resistance to woody species.

The species selected for this study showed significant differences in relation to phenolic compounds and tannin contents, which indicates that a homogeneous response of plants to fire could not be expected. Further, the high coefficients of variation on the concentration of secondary metabolites suggest a high intrapopulation variability playing a role in adaptation to changing environmental conditions [68,69]. The higher total phenolic compound contents have been observed in both Anacardiaceae species, *S. lorentzii* and, *S. johnstonii*. This family is also characterized by the occurrence of volatile compounds, which are related to plant flammability [9,11,12,70–72]. In addition to the above species, *S. mistol* (Rhamnaceae) showed the highest tannin contents. Therefore, the high interspecific variability observed in the phenolic compound and tannin contents could promote a

high flammability of native plant communities [68,69]. However, these mechanisms have been poorly conceptualized in the native species of the Chaco Region [11].

### 4.3. Resprouting Measurements

In this study, the post-fire survival was high since the fire intensity of the experimental burns was not severe enough to cause mortality in the selected species [33,34,73]. In our EB, we obtained basal resprouting, which is consistent with previous patterns observed in experimental burns in the Chaco Region [33,36]. Probably, the absence of epicormic resprouts could be the result of both the small size of plants and the thin bark (i.e., unprotected meristems) [36,73]. In this study, we found that fire intensity was relatively uniform, due to the absence of significant differences in burning time and flame length applied to the studied species. However, we found differences among species in burnt biomass, which suggests that fire severity could affect the post-fire plant responses observed in our study. These results confirm the role of the loss of aerial biomass in the resprouting vigor as a survival strategy to recover the vegetative structure [7,74]. We suggest that medium to high-intensity burns could promote basal resprouting, due to the loss of aboveground biomass inducing the formation of multi-stemmed shrubs and trees [51]. Similar results were reported by Herrero et al., (2015), who mentioned the predominance of basal resprouting in native woody species from the Chaco Serrano Region.

### 4.4. Biochemical Compounds Affecting the Resprouting Capacity

In contrast with our expectations, the phenolic compounds and tannins were the most important compounds regulating resprouting capacity. Possibly, this response could be related to the high radiation level and desiccation characterizing the postfire environments [75]. A similar process was reported by El Omari et al. (2013) which informed an increase in these contents in coincidence to the higher light availability and photosynthetic activity presented by resprouts as compared to control plants. This suggests that these compounds are involved in the detoxification mechanisms that enhance the resprouting protection and post-fire survival [28]. Additionally, these compounds play an important role in bud protection due to their antioxidant activity [76]. Even though we did not find a significant effect of both total chlorophylls and chlorophyll *b* on resprouting capacity, we found a significant effect of chlorophyll *a* on this capacity, which suggests a remarkable need to recover the photosynthesis efficiency in the post-fire environment. Considering that chlorophyll *a* and *b* are bioindicators of the plant physiological status, the maintenance of chlorophyll *b* instead of the requirements of chlorophyll *a* could suggest greater physiological stability of the studied species [15]. Chlorophyll *a* is the principal pigment in the photosynthesis process, whereas chlorophyll *b* is a subsidiary photosynthetic pigment that absorbs light in a different wavelength range to pass on to chlorophyll *a* [17].

### 4.5. Biochemical Compounds Affecting the Growth of Resprouts

Our results again showed the importance of the phenolic compounds in the resprouting response. These compounds had the most important effect on the number of resprouts, as well as in the diameter and height of resprouts. These results highlight the functional importance of the phenolic compounds in the plant response to environmental changes due to their antioxidant activity [13,76]. For example, a high concentration of these compounds could represent an ecological response to different forms of biotic and abiotic stress [77].

Regarding the shrubby species, we observed a complex biochemical response to fire which affected the growth of resprouts, whereas tree species were partially affected by the biochemical compounds studied. Among the tree species, we observed a lower effect of the biochemical response on the resprouts of *S. lorentzii* than in *A quebracho- blanco* which could be related to the dominance of the last-mentioned species in the disturbed forest of the region (Brassiolo 2005). These results highlight the role of the bioactive compounds in the resprouting. Additionally, our results seem to indicate that in the shrubby species the biochemical response could act as a driver of resprouting. In these

species, the photosynthetic pigments could enhance the availability of reserves for the growth of resprouts, and the increase in the secondary metabolites contributes to protecting the newly formed structures in the post-fire conditions [7,55,78,79].

*4.6. Association between the Biochemical Response to Fire and Resprouting*

Our results support our hypothesis that biochemical response to fire influences the resprouting. Indeed, we proposed a biochemical response index that allows us to compare the synergy effect of the bioactive compounds in the resprouting of each species. The significant correlation between both the resprouting capacity and the growth of resprouts with the biochemical response to fire suggests that bioactive compounds may have an important role in the post-fire vegetation recovery in our region, where these compounds could act as drivers of resprouting, mainly in shrubby species.

In our results, *A. emarginata, S. johnstonii*, and *C. ehrenbergiana* showed the higher resprouting capacity and biochemical response to fire, which could explain the capacity of these shrubby species to establish in strongly disturbed environments within the study area (Conti and Díaz 2013). In contrast, the tree species, *S. mistol, S. lorentzii*, and *A. quebracho-blanco*, showed the lowest values in RC and biochemical response to fire, in coincidence to the lower number, diameter, and height of resprouts. These results confirmed that resprouting in juveniles of tree species tends to diminish with plant height in low-intensity burns [34]. As reported by Bravo et al. [34,36], the studied species are characterized by having a low growth rate caused by their hard and heavy woods. Besides, the size of the bud bank differs among the studied species, and the resprouting ability decreases with the plant age (mainly in tree species) [36].

## 5. Conclusions

Our results support our hypothesis that experimental burns caused a considerable increment in the biochemical compounds studied within the time scale considered. Additionally, this biochemical response to fire was correlated to the post-fire resprouting, mainly in shrubby species. Our results indicate that phenolic compounds may have an important role in both resprouting capacity and in the growth of basal resprouts of the studied species, while photosynthetic pigments could play a significant but minor role in this process. As predicted, EB produced changes in the biosynthesis of active compounds, which are caused by physical stress and the time elapsed after the event. In our study region, these results seem to indicate that the photosynthetic process and recovery of the biomass of studied species may be insured under the natural conditions of our area. Besides, antioxidant compounds as phenolic and tannins showed an increase even 26 months after EB, indicating that their protective effect on metabolic processes is prolonged along time. However, responses observed after EB could not be related exclusively to fire, but also to a combination of drought, high UV-radiations and, pathogen responses typical of post-fire environments, which has been described by other authors as an exaptation. More quantitative studies about the physiological level responses are desirable to improve the understanding of the mechanisms involved in the resprouting and their effect on biomass production.

**Author Contributions:** Conceptualization, A.C.S.-G., M.A.N., and S.B.; Methodology, A.C.S.-G., F.d.C., and E.M.G.; data curation, A.C.S.-G. and F.d.C.; writing—original draft preparation, A.C.S.-G., S.B. and M.A.N.; writing—review and editing, S.B., M.A.N. and D.M.M.-T.; all authors discussed the results and commented on the manuscript. All authors have read and agreed to the published version of the manuscript.

**Funding:** The authors are grateful to the Universidad Nacional de Santiago del Estero (UNSE); Consejo Nacional de Investigaciones Científicas y Técnicas (CONICET) and the Ministry of Science, Technology and Productive Innovation. Besides, A.C.S.-G. acknowledges for her fellowship granted by CONICET.

**Institutional Review Board Statement:** Not applicable.

**Informed Consent Statement:** Not applicable.

**Data Availability Statement:** The data supporting the results in this paper are archived at the Institutional Repository of the National University of La Plata "SEDICI" (http://sedici.unlp.edu.ar/handle/10915/93714).

**Acknowledgments:** A.C.S.-G. acknowledges her doctoral fellowship granted by CONICET. We would like to thank the assistant staff and our field colleagues from Laboratorio de Antioxidantes y Procesos Oxidativos (LAPOx), FAyA, UNSE and Cátedra de Botánica General, FCF, UNSE.

**Conflicts of Interest:** The authors declare no conflict of interest.

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
