# Peer review of "How Do Plants Respond Biochemically to Fire? The Role of Photosynthetic Pigments and Secondary Metabolites in the Post-Fire Resprouting Response"

_forests, doi:10.3390/f12010056_

Round 1

Reviewer 1 Report

Aspects to consider:

The introduction from line 57 to line 89 is interesting but could be written in a briefer way, given that there is some information that appears repeatedly.

Table 1. Separate in three different tables.

  1. Species characteristics
  2. Burn characteristics. Why is only the 2016 burn included? All carried out burns should be included. Should be moved to part 2.2. Characterization of experimental burns.
  3. The resprouting capacity table should be placed in the results (3.3.)

Line 290. Figure 1. It is not necessary to explain the method used to carry out the study (“Samples…Eb 2016”), since this is previously explained in methods.

Line 360. Figure 3. As in previous figures, it is not necessary to add methodology aspects below the figure.

Figure 3A, results of phenolic compounds of all species are shown in control and in the different burns but there is too much heterogeneity in the different species figure (3B). How can these differences be explained?

The same differences can be found in figure 3C for ECT.

Line 443. I believe there is a mistake: Figure 2016?

Line 501. The bibliographic citation lacks a number: Jaureguiberry and Diaz (2015).

Author Response

Reply to Referee Report
We thank the editor in chief, the assistant editor, and referees for their careful reading of our manuscript and for their additional comments and suggestions. In the updated version, we have addressed all of these. Below, we present our response to the referee’s points and a description of the changes introduced in the text:

The introduction from line 57 to line 89 is interesting but could be written in a briefer way, given that there is some information that appears repeatedly.
We accepted the observation and summarize the introduction. However, it was difficult to summarize some lines due to the importance of their content.

Table 1. Separate in three different tables.
1. Species characteristics
2. Burn characteristics. Why is only the 2016 burn included? All carried out burns should
be included. Should be moved to part 2.2. Characterization of experimental burns.
3. The resprouting capacity table should be placed in the results (3.3.)
Suggestion was accepted. We would like to thank the reviewer in advance for this
observation. Even though we performed and collected the data from three experimental burns, in the table 3 we clarify that resprouting measurements were made in the burn plants from EB 2016.

Line 290. Figure 1. It is not necessary to explain the method used to carry out the study (“Samples…Eb 2016”), since this is previously explained in methods.
We accepted the observation.

Line 360. Figure 3. As in previous figures, it is not necessary to add methodology aspects below the figure.
We accepted the observation.

Figure 3A, results of phenolic compounds of all species are shown in control and in the different burns but there is too much heterogeneity in the different species figure (3B). How can these differences be explained?
The same differences can be found in figure 3C for ECT.
As this research was conducted in an experimental site with native species, a high
interspecific and intraspecific variation could be expected. Indeed, due to the function of the secondary metabolites, if a species or population shows individual variability in the
concentration of these compounds, it would imply a better response to environmental changes, which suggest a better response to environmental heterogeneity. A high intrapopulation variability could show the ability of the species or population to adapt to the biotic and abiotic variations in the environment (Valares 2011). At lines 503-507 we have tried to explain the effect of the species variability in the plant response to fire: The species
selected to this study showed significant differences in relation to phenolic compounds and tannin contents, which indicates that a homogeneous response of plants to fire could not be expected. Further, the high coefficients of variation on the concentration of secondary metabolites suggest a high intrapopulation variability playing a role in adaptation to changing environmental conditions
[68,69]”

Line 443. I believe there is a mistake: Figure 2016?
Thank you very much for your careful review. We were mentioned the burnt plants in 2016 and we corrected the mistake.

Line 501. The bibliographic citation lacks a number: Jaureguiberry and Diaz (2015).
Thank you for your observation. We corrected it.
_________________________________________________________________

Each suggestion or recommendation performed to clarify the manuscript, was considered, reviewed and attend. The changes in the updated version were highlighted within the document by using the track changes mode in MS Word.
We truly thank the editors and reviewers for devoting time to read the paper and suggest modifications that improved our presentation. We hope that the paper is now suitable for publication.

Best,
The authors

Reviewer 2 Report

 This paper reports the effect of experimental burns on biochemical traits, including photosynthetic pigments and antioxidants in the Chaco region of Argentina. The relationship of these traits to post-fire behavior such as resprouting is not often reported, therefore this is a novel and important study. The authors have done a comprehensive analysis of resprouting traits and biochemical compounds across three years of experimental burns and the results are presented nicely.

I have some comments that I hope will improve the paper:

  • I think the colors used in the figures could be improved, as currently the difference in color doesn’t show up when viewed in greyscale.
  • The authors often state ‘a non-significant decrease’. The authors often discuss non-significant results with the same level of emphasis as the significant results (e.g. line 265-280), I think some of the results could be re-written to emphasise the significant results more first, then comment on potential trends in that were non-significant.
  • Line 28: Change ‘As results’ to ‘our results demonstrate’
  • Line 32: I think it’s better to say ‘minor but significant’ rather than ‘significant but minor’
  • Line 42: Change ‘Fire has always been an important…’ to ‘Fire has always played an important…’
  • Line 47: Reword ‘becoming in increased severity’ (e.g. to just ‘increasing in severity’)
  • Line 58: Change ‘Plants have been’ to ‘Plants are’
  • Line 59-60: Include some examples of compounds that are promoted by fire from the literature
  • Line 66: Change ‘due to’ to ‘as’
  • Line 72: This sentence could be made more concise by changing ‘Regarding to carotenoids, those compounds are considered…’ to simply ‘Carotenoids are considered….’
  • Line 90: I think this sentence could be reworded more clearly.
  • Line 98: (typo) missing the word ‘such’ after ‘disturbances’
  • Line 233: As you have collected data on mortality it would be nice to include this in your analysis, it could just be added to a table. Did it differ between the species?
  • Line 246-262: Was this data from 6 or 12 month samples after that experimental burn?
  • Line 282: Is this difference expected due to seasonal changes?
  • Figure 1: The color scheme for this figure could be improved as it is impossible to tell the colors apart in greyscale (this also applies to figure 2).
  • Figure 1: It’s good that you have included the statistical analysis in the figure, but is it necessary to include comparisons of everything? Do you need to comparisons between species, it would be clearer if you just focused on the differences between Control and Burned for each species separately (also applicable to figure 2).
  • Line 308: Again, is this difference definitely due to the elapsed time, or is it because of the time of year/season?
  • Line 333: Do you mean micrograms, rather than milligrams here? Otherwise you have more phenolic compounds than original sample!
  • Line 351: It’s unclear what you mean by ‘tendency related to foliar persistance’?
  • Table 2: I think it would be valuable to include the non-significant values too in all the tables, but have the significant ones in bold.
  • Line 405: (typo) Correct ‘that’ to ‘than’
  • Line 511: It would be useful to explain exactly how phenolics and tannins promote flammability.
  • Line 536: I would like to see some more discussion of the specific role of chlorophyll a vs chlorophyll b so you could hypothesize why the responses differ.
  • Line 567: In this section I think it would be useful to include some information from the literature about the relative growth rates of your studied species. Are the normal growth rates the same or correlated to the resprouting growth rates?
  • Line 579: Are some of the differences you observe in biochemical analyses due to comparing new leaves (in the burned trees) to old leaves in the control trees? Have you controlled for leaf age? Or have you compared the traits in old vs young control leaves?

Author Response

Reply to Referee Report

We thank the editor in chief, the assistant editor, and referees for their careful reading of our manuscript and for their additional comments and suggestions. In the updated version, we have addressed all of these. Below, we present our response to the referee’s points and a description of the changes introduced in the text:

This paper reports the effect of experimental burns on biochemical traits, including photosynthetic pigments and antioxidants in the Chaco region of Argentina. The relationship of these traits to post-fire behavior such as resprouting is not often reported, therefore this is a novel and important study. The authors have done a comprehensive analysis of resprouting traits and biochemical compounds across three years of experimental burns and the results are presented nicely.
We would like to thank the reviewer for his/her positive comments on the manuscript. Below is our response to his/her observations.

I think the colors used in the figures could be improved, as currently the difference in color doesn’t show up when viewed in greyscale.
Suggestion was accepted. The figure colors were changed.

The authors often state ‘a non-significant decrease’. The authors often discuss nonsignificant results with the same level of emphasis as the significant results (e.g. line 265-280), I think some of the results could be re-written to emphasise the significant results more first, then comment on potential trends in that were non-significant.
We would like to thank the reviewer for this observation. However, we presented the results in this order following the objectives that we proposed. As our main objective was to evaluate the plant response to fire, we considered important to present these results at the beginning of each section regardless of the significance or not significance of the data. However, if you continue considering that we have to re-write the results, we will accept the suggestion and we will modify them.

Line 28: Change ‘As results’ to ‘our results demonstrate’
Suggestion was accepted.

Line 32: I think it’s better to say ‘minor but significant’ rather than ‘significant but minor’
Suggestion was accepted.

Line 42: Change ‘Fire has always been an important…’ to ‘Fire has always played an important…’

Suggestion was accepted.

Line 47: Reword ‘becoming in increased severity’ (e.g. to just ‘increasing in severity’)
Suggestion was accepted.

Line 58: Change ‘Plants have been’ to ‘Plants are’
Suggestion was accepted.

Line 59-60: Include some examples of compounds that are promoted by fire from the literature
Suggestion was accepted, we included the biosynthesis of terpenoids.

•Line 66: Change ‘due to’ to ‘as’
Suggestion was accepted.

Line 72: This sentence could be made more concise by changing ‘Regarding to carotenoids, those compounds are considered…’ to simply ‘Carotenoids are considered….
Suggestion was accepted.

Line 90: I think this sentence could be reworded more clearly.
Suggestion was accepted. We reorganize the phrase at line 85: “
This work proposed a novel approach of the resprouting since a biochemical understanding

Line 98: (typo) missing the word ‘such’ after ‘disturbances’
Suggestion was accepted.

Line 233: As you have collected data on mortality it would be nice to include this in your analysis, it could just be added to a table. Did it differ between the species?
We did not report the mortality since the fuel load used in the experimental burns was not severe enough to cause mortality in the selected species. We used a low fine fuel load (4000 kg DM ha
-1) which correspond to the aboveground biomass in the Chaco Region. However, as fire is a common disturbance in the region, species has developed strategies to survive. (Ledesma et al. 2018, Bravo et al. 2014). This fire intensity was enough to affect the studied plants and allow us to observe a plant response to fire. In the updated version of the manuscript, we incorporated this variable in the table 3. Besides at line 364 and line 515 we described the results and discussion of this variable.

Line 246-262: Was this data from 6 or 12 months samples after that experimental burn?
This data was for samples collected 6 months after EB in coincidence with the same period of the resprouting measurements. At lines 257-258 this observation was incorporated.

Line 282: Is this difference expected due to seasonal changes?
•Line 308: Again, is this difference definitely due to the elapsed time, or is it because of the time of year/season?
According to our experimental design and our results, we could affirm that the significant differences in the photosynthetic pigments concentration are not mainly caused by the fire effect but rather by a time effect since the experimental burn. However, our experimental design does not allow us to identify whether the observed changes were caused by the elapsed time, or by the time of year / season, due to the fact that we do not continue the measurements in the time. The observed differences in the species according to their foliar persistence could suggest an important effect of seasonality on the biosynthesis of these
compounds which also has been reported by other studies (Takashima et al. 2004)

Figure 1: The color scheme for this figure could be improved as it is impossible to tell the colors apart in greyscale (this also applies to figure 2).
Suggestion was accepted. We changed the colors of the figures.

•Figure 1: It’s good that you have included the statistical analysis in the figure, but is it necessary to include comparisons of everything? Do you need to comparisons between species, it would be clearer if you just focused on the differences between Control and Burned for each species separately (also applicable to figure 2).
We included figures 1B and 2B due to the results of the statistical analysis for photosynthetic pigments. As our results did not showed a significant effect of fire but rather a significant effect of both the time elapsed after EB and the species, we considered important to show these results discriminating among control and burnt plants for each species. These figures allowed us to explain the observed differential behavior related to the foliar persistence as perennial and deciduous species showed different fire response tendency at 12 months after EB. Additionally, we decided to incorporate these comparisons in a single figure to pretend to show the results concise. However, as we mentioned before, if you continue considering that these figures are not appropriate, we will accept the suggestion and we will modify them.

Line 333: Do you mean micrograms, rather than milligrams here? Otherwise you have more phenolic compounds than original sample!
We would like to thank the referee for this observation. We had made a typing error writing the manuscript. The correct unit for the content of phenolic compounds is mg of gallic acid
1000 mg- 1 of sample.

Line 351: It’s unclear what you mean by ‘tendency related to foliar persistance’?
We have tried to clarify the meaning of this sentence (L341-342):
“The results were not related to the foliar persistence of the studied species”.

Table 2: I think it would be valuable to include the non-significant values too in all the tables, but have the significant ones in bold.
We accepted the observation, and we incorporated the non-significant values in all the tables.

Line 511: It would be useful to explain exactly how phenolics and tannins promote flammability.
Even though the antecedents reported that phenolic compounds could promote the flammability by reducing ignition temperatures in both foliage and litter (Bowman et al. 2014), the underlying mechanisms in this process have been poorly conceptualized. We incorporated this sentence at the end of the paragraph: “
However, these mechanisms have been poorly conceptualized in the native species of the Chaco Region (Bowman et al. 2014).” (L513-514)

•Line 536: I would like to see some more discussion of the specific role of chlorophyll a vs chlorophyll b so you could hypothesize why the responses differ.
Suggestion was accepted. At line we incorporated the role of chlorophyll a and b at line 541-546:
“Considering that chlorophyll a and b are bioindicators of the plant physiological status, the maintenance of chlorophyll b instead of the requirements of chlorophyll a could suggest a greater physiological stability of the studied species [15]. Chlorophyll a is the principal pigment in photosynthesis process, whereas chlorophyll b is a subsidiary photosynthetic pigment that absorbs light in a different wavelength range to pass on to chlorophyll a [17].”

Line 567: In this section I think it would be useful to include some information from the literature about the relative growth rates of your studied species. Are the normal growth rates the same or correlated to the resprouting growth rates?
We accepted the suggestion. The growth rates observed in our study correspond to the normal growth rates of the studied species according to previous studies (Bravo et al. 2014, 2018, 2019)
. At line 578-581 we incorporated: “As reported Bravo et al. [34,36] the studied species are characterized by having a low growth rate caused by their hard and heavy woods. Besides, the size of the bud bank differs among the studied species, and the resprouting ability
decreases with the plant age (mainly in tree species) [36]. “

Line 579: Are some of the differences you observe in biochemical analyses due to comparing new leaves (in the burned trees) to old leaves in the control trees? Have you controlled for leaf age? Or have you compared the traits in old vs young control leaves?
Thank you for your observation. As you mentioned, there are some differences in the content of photosynthetic pigments and phenolic compounds according to the leaf age. Young leaves show greater biosynthesis of these metabolites as compared to mature leaves and stems (Valares et al. 2016). For this reason, in our study we tried to select leaves of the same
age. However, since our main aim was to evaluate the fire effect on the plant response, in some cases this was not completely possible. Consequently, for these cases, we selected resprouting leaves in burnt plants and young leaves in control plants that did not have resproutings yet, to intent to standardize the measurements.
_________________________________________________________________

Each suggestion or recommendation performed to clarify the sentence meaning or to correct a grammatical mistake in the manuscript, was considered, reviewed and attend. The changes in the updated version were highlighted within the document by using the track changes mode in MS Word, however, the number of the lines used in this document as reference to show a correction within the updated manuscript, were taken from the accepted changes version.
We truly thank the editors and reviewers for devoting time to read the paper and suggest modifications that improved our presentation. We hope that the paper is now suitable for publication.

Best,
The authors
